# The Effects of Music-Based Interventions for Pain and Anxiety Management during Vaginal Labour and Caesarean Delivery: A Systematic Review and Narrative Synthesis of Randomised Controlled Trials

**DOI:** 10.3390/ijerph20237120

**Published:** 2023-11-29

**Authors:** Amy Rose Hunter, Annie Heiderscheit, Megan Galbally, Davide Gravina, Hiba Mutwalli, Hubertus Himmerich

**Affiliations:** 1Centre for Research in Eating and Weight Disorders (CREW), Department of Psychological Medicine, Institute of Psychiatry, Psychology and Neuroscience, King’s College London, London SE5 8AF, UK; amy.hunter@kcl.ac.uk (A.R.H.); davide.gravina@kcl.ac.uk (D.G.); hiba.mutwalli@kcl.ac.uk (H.M.); 2Mental Health Studies Programme, Institute of Psychiatry, Psychology & Neuroscience, King’s College London, London SE5 8AB, UK; 3Cambridge Institute for Music Therapy Research, Anglia Ruskin University, Cambridge CB1 2LZ, UK; annie.heiderscheit@aru.ac.uk; 4School of Clinical Sciences, Department of Psychiatry, Monash University, Clayton, VIC 3168, Australia; megan.galbally@monash.edu; 5Department of Clinical and Experimental Medicine, University of Pisa, 56127 Pisa, Italy; 6Department of Clinical Nutrition, College of Applied Medical Sciences, Imam Abdulrahman Bin Faisal University, Dammam 31441, Saudi Arabia; 7South London and Maudsley NHS Foundation Trust, Bethlem Royal Hospital, Monks Orchard Road, Beckenham BR3 3BX, UK

**Keywords:** music, music therapy, labour, caesarean section, pain, anxiety

## Abstract

Music-based interventions are not physically invasive, they usually have minimal side effects, and they are increasingly being implemented during the birthing process for pain and anxiety relief. The aim of this systematic review is to summarise and evaluate published, randomised controlled trials (RCTs) assessing the effects of music-based interventions for pain and anxiety management during vaginal labour and caesarean delivery. Following the PRISMA guidelines, a systematic search of the literature was conducted using: PsychInfo (Ovid), PubMed, and Web of Science. Studies were included in the review if they were RCTs that assessed the effects of music on pain and anxiety during vaginal and caesarean delivery by human mothers. A narrative synthesis was conducted on 28 identified studies with a total of 2835 participants. Most, but not all, of the included studies assessing music-based interventions resulted in reduced anxiety and pain during vaginal and caesarean delivery. Music as part of a comprehensive treatment strategy, participant-selected music, music coupled with another therapy, and relaxing/instrumental music was specifically useful for reducing light to moderate pain and anxiety. Music-based interventions show promising effects in mitigating pain and anxiety in women during labour. However, the long-term effects of these interventions are unclear.

## 1. Introduction

### 1.1. Labour Pain and Anxiety

Childbirth is painful, and the pain usually begins during the latent stage of vaginal labour (the period from when the mother’s cervix begins to soften and dilate) and continues to intensify during the established phase (where cervical dilation has reached 4 cm), up until the child is born [1]. Whilst labour pain is a multifaceted and subjective phenomenon, for most women it is the most extreme form of pain that they will endure and may lead to negative physiological and psychological outcomes [2,3]. Excessive increases in physiological factors, including the mother’s heart rate (HR), blood pressure (BP), oxygen intake, and respiratory rate (RR) have been shown to have negative effects on the mother and unborn child [3,4]. Consequently, unbearable, unalleviated labour pain can lead to extreme exhaustion, mental distress, anxiety, and cognitive dysfunction in the mother [3]. Anxiety and distress activate the sympathetic nervous system by increasing the secretion of hormones such as adrenaline and noradrenaline into the bloodstream, which can obstruct oxytocin secretion [5]. Consequently, contractions may change in pattern, decrease in rate, or discontinue entirely [1], thus affecting labour progression and increasing the risk of an emergency caesarean section (C-section) being performed [6,7]. These hormonal changes can also reduce blood supply to the uterus, therefore impacting foetal oxygen levels and putting the unborn child in danger [1,8]. While labour pain is a crucial indicator of childbirth, it should be effectively managed to prevent obstetric difficulties and the need for further medical interventions.

### 1.2. Caesarean Section and Anxiety

An alternative method for childbirth is a C-section i.e., the surgical removal of the baby and placenta from the mother’s abdominal and uterine wall [9]. Women undergoing C-sections are likely to experience elevated levels of psychological anxiety, as despite it being a common surgical procedure, there is a higher risk in comparison to vaginal delivery due to the potential complications, including blood loss, laceration infection, endometriosis, venous thromboembolism, collapsed lungs, and thrombophlebitis [9,10]. Perioperative anxiety can be defined as a distressing and unpleasant feeling causing worry and nervousness throughout the preoperative, operative, and postoperative period that creates an emotional reaction to a possible threat [11,12]. Perioperative anxiety has physiological implications as the sympathetic adrenal–medullary system becomes aroused, which in turn may affect the circulatory system, leading to an increased risk of complications, including constriction of the mother’s coronary arteries, greater blood viscosity, and increased risk of a heart attack [13]. It is therefore crucial that heightened anxiety is detected and managed throughout the perioperative period, as an inability to do so may delay recovery, increase hospital admission length, and intensify maternal pain sensitivity, leading to a greater demand for pain-relieving drugs [14,15]. Research by Wyatt et al. [16] revealed that a substantial proportion of anxious women took benzodiazepines pre-surgery to manage their anxiety despite knowing the negative side effects associated with the drug, for example, respiratory depression, indicating a need to consider alternative forms of anxiety management with greater risk aversion for women undergoing C-sections.

### 1.3. Music-Based Interventions for Pain and Anxiety Management

The management of pain and anxiety during vaginal labour and C-sections is a critical worry for the mother and healthcare professionals [17]. The most common and successful methods of pharmacological pain relief during childbirth include epidural analgesia, nitrous oxide, and intravenous opioids [18]. However, these methods come with unwanted potential side effects. These include affecting the mother’s sensations of control, hindering labour progression, increasing the probability of additional interventions, including C-section [19], inducing drowsiness, and impacting the mother’s ability to safely breastfeed her baby [19]. Delivering alternative non-pharmacological treatments that grant the mother independence and active choice over pain management during childbirth can in turn lower anxiety and fear [20].

Music-based interventions are methods of non-pharmacological pain relief that have received increased interest in recent years [8]. Music is ever-present, emotive, social, and in its most elective form, occurs in every culture [21]. The history of music and its therapeutic role within the medical field has been discussed and reported on as far back as 4000 BC [21]. A variety of music-based therapies, strategies, and methods may be beneficial for promoting health and well-being [22].

Many scientific articles report the therapeutic value of music-based interventions during childbirth [23,24,25,26]. Music can positively affect the physiology of mothers during labour by activating the primary auditory cortex, which further stimulates the limbic system, brain stem, hypothalamus, and cerebral cortex. As the auditory cortex and the pain centre of the cerebral cortex are neighbouring, and thus highly connected, music can activate endorphin secretion, increase oxygenation in organs and tissues, and reduce pain sensitivity [27].

Previous systematic reviews have already summarised the effect of music on pain and anxiety during labour and C-section [28,29,30,31]. However, two of these reviews included studies with multiple research designs e.g., quasi-experimental designs, with a risk of potential bias due to the absence of randomisation [28,29]. One systematic review examined the effect of music on anxiety in mothers during C-sections only [31] and another assessed the use of music exclusively for anxiety during labour [30]. As the use of non-pharmacological interventions continues to gain popularity [8], and novel studies have recently been published, an updated systematic review seems timely.

To the best of our knowledge, no systematic review has specifically focused on randomised controlled trials (RCTs) that tested the effects of music-based interventions for anxiety and pain management during both vaginal labour and caesarean sections. RCTs are considered the gold standard for effective research, as randomisation limits bias, allowing researchers to establish cause-and-effect relationships between an intervention and an outcome [32]. Since the therapeutic use of music and music therapy are considered a physically non-invasive, generally cost-effective, natural intervention with limited side effects [8,33], policymakers and healthcare officials should seriously consider its place during childbirth so that women can make educated and informed decisions regarding their birth preferences using the current evidence available.

### 1.4. Aims and Objectives

This systematic review aims to summarise and evaluate the available literature on the effects of music-based interventions for pain and anxiety management during vaginal labour and caesarean delivery.

## 2. Materials and Methods

### 2.1. Search Strategy

The present systematic review was conducted in line with the Preferred Reporting Items for Systematic Reviews and Meta-Analyses (PRISMA) guidelines [34]. Appendix A shows the completed PRISMA checklist for this systematic review. After preliminary scoping searches in March 2023, PsychInfo (Ovid), PubMed, and Web of Science were selected to obtain studies for this review.

The general search terms were:Birth, labour, childbirthCaesarean birth, caesarean sectionMusic intervention, music therapy

However, we made adaptations for these general search terms according to the requirements of each electronic database as explained below.

The search criteria for each electronic database used the appropriate keywords and MeSH (Medical Subject Headings) for labour and music-based interventions. Boolean operators, truncation, wildcards, and proximity searches were included to increase the sensitivity and comprehensiveness of the search [35]. No limits were applied to the search. Due to extensive database coverage, no additional hand-searching was conducted. The following search terms were used: PsychInfo (Ovid): birth/or caesarean birth/or “labor (childbirth)”/or natural childbirth/or birth or caesarean birth or caesarean section or cesarean birth or cesarean section or C-section or childbirth AND music intervention* or music-based intervention* or music therap* or music therapy/or (effect* adj2 music-based intervention*) or (effect* adj2 music intervention*) or (effect* adj2 music therap*); Web of Science: (ALL = (birth* or childbirth or labo$r or “c*sarean birth” or “c*sarean section” or “c-section” or “natural birth” or parturition)) AND ALL = (“music-based intervention*” OR “music therap*” OR “music intervention”); PubMed: (“Parturition”[Mesh] OR “Natural Childbirth”[Mesh] OR “Cesarean Section”[Mesh] OR birth[Title/Abstract] OR natural birth*[Title/Abstract] OR labour[Title/Abstract] OR labor[Title/Abstract] OR childbirth[Title/Abstract] OR “caesarean section” OR “caesarean birth” OR “cesarean birth” OR C-section) AND (“music-based intervention*”[Title/Abstract] OR “music intervention*”[Title/Abstract] OR “music therap*”[Title/Abstract] OR “Music Therapy”[Mesh]).

### 2.2. Eligibility Criteria

The eligibility criteria were created and confirmed based on published guidance how to write systematic reviews [36]. This included the following:

#### 2.2.1. Inclusion Criteria

Participants were human.Only Randomised controlled trials were used.Studies were on the effect of music or music-based interventions on pain management or maternal well-being during childbirth, including vaginal birth and caesarean section.Music or music therapy was used as an intervention.Studies were published in English or German.

#### 2.2.2. Exclusion Criteria

Animal studies.Case reports.Review articles.Perspective Papers.Letters.Master or doctoral theses.Systematic reviews or meta-analysesArticles that do not report data on mothers during labour.No music applied.Meeting abstracts.Findings that failed to report results/clinical outcomes.

### 2.3. Study Selection and Risk of Bias

The title and abstracts of studies obtained through the searches were imported into EndNote and duplicates were removed using the software. Articles were then imported into Rayyan to detect and remove further duplications.

A two-step screening method involving a preliminary title and abstract screening, followed by a thorough full-text screening was implemented. Two independent reviewers (ARH and DG) screened the title and abstracts against the prespecified eligibility criteria outlined above. During the second stage, the first reviewer hand-selected significant information from each included paper, including bibliographic information, sociodemographic information, study design, measurement instruments, types of interventions, main results, and statistical significance. Both reviewers (ARH and DG) separately used this information to complete a quality assessment of each study using the Scottish Intercollegiate Guidelines Network (SIGN) (See Appendix A). Any inter-observer discrepancies between reviewers during the screening process were resolved at the ultimate stage via reviewer consensus.

### 2.4. Analysis

To categorise, analyse, and compare the studies involved in this systematic review, guidance for narrative synthesis by Popay et al. was followed [37]. The papers included in this review were summarised and ordered into groups in keeping with the theme.

## 3. Results

Applying the search methods outlined above, a sum of 501 studies was extracted. This was reduced to 351 once duplicates were removed. The PRISMA Flow Diagram, illustrating the progression of study selection, is shown in Figure 1 [34]. A detailed summary of all included studies (*n* = 28), the specific methods they used to measure pain and anxiety, and the relevant results is shown in Table 1. Table 2 shows a more condensed and readily understandable synopsis focussing on the overarching main findings.

### 3.1. Studies Examining the Effects of Music-Based Interventions during Vaginal Labour

#### 3.1.1. Music-Based Intervention as the Experimental Group Compared with a Control Group

In two studies, participants chose the music played during delivery from a predetermined list. Phumdoung and Good [3] studied the effects of listening to relaxing music compared with traditional childbirth, with no music played during the active phase of labour, on the sensation and distress of pain in 110 primiparous women. The experimental group experienced less pain sensation and discomfort compared with the control group. Music only partially inhibited the increase in pain over time as the distress of pain did not increase significantly throughout the first hour in the experimental group; however, both groups displayed significant differences in sensation and distress over time. In both groups, the distress of pain values were constantly rated lower than the sensation, showing a variable effect of music on stress reduction in labouring mothers.

Research by Liu et al. [38] examined the effects of relaxing music in comparison to routine care with no music intervention on pain and anxiety in 60 primiparous mothers during the latent and active phases of labour. Listening to music throughout the latent phase of labour was shown to decrease pain and anxiety in mothers. However, these effects were not significant throughout the active phase, indicating that the impact of music may be more evident in the early stages of labour.

In three studies, participants selected the songs that were played in the music intervention group from a predetermined list of music from different genres. Simavli et al. [39] assessed the effect of music on postpartum pain, anxiety, depression, and satisfaction compared with standard prenatal care in 161 women. The results showed significant decreases in minor and major depression rates between the experimental and control groups on the first and eighth days postpartum. Additionally, the participant satisfaction rate significantly increased in the music group compared with the control group at 2, 12, and 24 h postpartum. Postpartum pain and anxiety were also significantly lower in the music group compared with the control group at all the time intervals.

Another study by Simavli et al. [42] examined the effect of music on labour pain, anxiety, maternal hemodynamics, foetal neonatal factors, and postpartum pain medication requirements in 156 first-time mothers. The results relevant to the current review revealed that during the first and second phases of labour and postpartum, pain and anxiety in women in the music group were rated lower in comparison to women who received routine obstetric care with no music. Additionally, objective measurements, including diastolic blood pressure (DBP), systolic blood pressure (SBP), and HR, were significantly lower during labour and postpartum in the music group. Labour duration was shorter in the music group during the active phase and the second stage, indicating physical relaxation. Finally, participants in the music group made fewer requests for postpartum pain-relieving medication; however, this was only assessed during the early stages (8 and 24 h) of postpartum, therefore its longer-term effects are unclear.

In a small study of 20 participants, Browning [45] studied the effectiveness of listening to anxiety-relieving music along with breathing techniques and progressive muscle relaxation (PMR) on increasing relaxation, feelings of personal control, and birthing satisfaction during labour. The mothers’ experiences of pain and the amount/rate of pain-relieving medication were also examined. Participants in the control group were taught breathing techniques and PMR but with no music. The findings indicated that women who received MT during childbirth experienced greater relaxation, an increased sense of personal control, and more positive feelings associated with their labour. However, MT did not significantly reduce the overall probability of requesting pain-relieving medication nor the quantity of pain-relief medication used.

Four RCTs used music-based interventions where the music was preselected by the researchers. Two studies, both conducted by the same author and each involving 409 participants, analysed the effectiveness of a specific musical piece, “musical journey through pregnancy” by Gabriel F. Federico, played prenatally during the birthing process [43,46]. García-González [43] found that the average length of the first phase of labour in the music intervention group was significantly shorter (4 h 36 min) than in the control group, which received no music (5 h 54 min). Additionally, women in the music group experienced more spontaneous and less medicated deliveries compared with women in the control group. Despite a lower proportion of C-sections being performed in the music group, this difference was not statistically significant.

García-González [46] examined state-trait anxiety in pregnant women in the music and control groups whilst considering factors associated with the birthing process. They observed that mothers in the music group who came to the hospital with a ruptured amniotic sac experienced significantly less anxiety compared with mothers in the control group who also arrived at the hospital with a ruptured amniotic sac. Participants in the music group underwent significantly fewer induced labours, C-sections, and episiotomies compared with those in the control group with no music.

Surucu et al. [40] assessed the effects of listening to Acemasiran mode music for 3 h during the active phase of labour compared with undergoing traditional labour with no music on pain and anxiety in 50 primiparous women. The mean pain value in the music and control groups were similar at the start and 30 min into labour. However, statistically significant differences in pain scores were found at hourly time points from 1 h to 8 h. State-anxiety was significantly lower in the music group compared with the control group. Additionally, women in the experimental group regarded their deliveries as easier and had longer contractions and faster labour progression compared with women in the control group.

A small study consisting of 30 participants examined the effects of reed flute music on pain and anxiety during the active phase of labour in pregnant women compared with standard care with no music intervention [44]. Women in the music group experienced lower levels of pain and anxiety compared with women in the control group.

One small study involving 30 participants permitted participants to choose the music genre/individual songs played during labour. Buglione et al. [41] explored the effects of music-based interventions on pain and anxiety in nulliparous women compared with standard labour and delivery care with no music intervention. Women experienced diminished levels of pain in the experimental group compared with the control group. Listening to music during labour and delivery was correlated with less pain 1 h postpartum and with lower levels of anxiety throughout the active phase, the second stage of labour, and 1 h postpartum. Differences in pain and anxiety levels between the music and control groups were not statistically significant 24 h and 48 h postpartum. Anxiety was higher in the music group than in the control group 48 h postpartum.

#### 3.1.2. Music-Based Interventions Compared with Other Therapies

Two studies analysed the effects of music-based interventions against other therapies where music was selected from a predetermined list by participants. Taghinejad et al. [49] compared the effects of massage versus listening to a choice of five types of traditional Iranian music on labour pain in 101 participants. The researchers found that massage and music were effective therapies for mitigating labour pain; however, participants in the massage therapy group experienced less pain than those in the music group. This distinction was most pronounced during the most excruciating pain phase, where massage therapy appeared more effective.

Dehcheshmeh and Rafiei [50] recruited 112 participants to either the music group where participants could listen to piano music or wave sounds for 30 min, Hoku point ice massage (HPIM), or standard labour care. Participants in both the music and HPIM therapy groups had significantly lower levels of pain compared with those in the control group from the start of the active phase of labour up to 4, 6, and 8 cm dilated. Although the effects of both music and HPIM were statistically comparable, the average value of pain intensity in the music group was lower than in the HPIM group.

Wan and Wen [51] investigated whether acupressure, listening to music, and a combination of both were effective at reducing pain in 238 primiparous mothers during the active phase of labour compared with a control group that received no intervention. Participants were recommended a range of music styles; however, it is unclear whether participants had a say in what was played. Anxiety scores in the acupressure and music groups were statistically lower than in the control group. The combination group also displayed significantly less anxiety than the control group from 1 h to 24 h; however, anxiety rates in the music group were markedly higher than in the acupressure and combination groups. Participants in all three experimental groups experienced less pain compared with participants in the control group. However, pain at 1, 4, and 8 h was statistically lower in the music-based intervention group than in the acupressure and combination groups. Birth satisfaction rates in the three experimental groups were significantly higher than in the control group, and there were no significant differences between the three experimental groups.

Estrella-Juarez et al. [48] compared labour and delivery outcomes between three groups: music, VR (involving images and relaxing sounds of the sea), and control (no intervention). The song musical journey through pregnancy by Gabriel F. Federico was preselected by the researchers for the music intervention. The results showed that the length of the first phase of labour was significantly shorter in both the music and VR groups compared with the control group. There were more spontaneous vaginal deliveries in the VR group (82.4%) than in the music and control groups (48.1% and 51.8%, respectively). The VR and music groups had statistically fewer episiotomies than the control group.

One RCT with a sample of 99 participants compared listening to music with a combination of dance/listening to music and a control [52]. For music, participants could choose three songs significant to them. Intergroup comparisons showed that the experimental groups experienced significant reductions in subjective ratings of pain and fear during the active phase of labour compared with the control group. Pain and fear scores in the control group were significantly higher than in the music and dance and music groups. Although both the music and music/dance groups showed statistically significant reductions in the perceptions of pain and fear, the mean values revealed that the dance and music intervention was slightly more effective than music only.

A Pilot RCT consisting of 90 participants studied the effects of massage therapy with relaxation methods versus music intervention with relaxation techniques and controls who received no music but were encouraged to attend standard antenatal classes [47]. The findings indicated that there were no significant differences in the subjective ratings of pain throughout childbirth between all three groups and, thus massage appears to be better or similar to music at providing pain relief during childbirth. Similarly, there were no differences in the use of analgesic medication between all groups. Psychological assessments post-birth indicated a tendency of participants in the massage and music groups to hold more favourable perspectives on childbirth, readiness, and a sense of personal control in comparison to participants in the control group.

#### 3.1.3. Music-Based Interventions as Part of a Larger Therapy

Two papers examined the effects of music as part of a larger intervention/therapy. Guo et al. [27] conducted a large study of 440 participants. The music group listened to relaxing/hypnotic music during the first phase of labour, intense rhythmic music during the late stage of the first phase of labour, and parent–child music during the second/third phases of childbirth. The control group received standard care with no music. The findings showed that music along with free-position delivery may result in decreased pain perception, pain tolerance, and overall pain count. The experimental group also experienced a longer first phase and total stage of labour; however, no statistical difference was evident in the second and third phases. Maternal haemorrhaging 2 h post-delivery was significantly less in the experimental group but the perineal score was higher compared with the traditional delivery group with no intervention. Post-delivery Apgar scores (infant HR, reflex reaction, muscle tone, and skin tone) were not significant between groups.

Perković et al. [53] examined the relationship between prenatal therapy consisting of childbirth education, breathing techniques, and classical music on pain perception and psychological symptoms during labour in 175 Iranian women. The findings indicated that pain perception was lower in mothers in the therapy group compared with those in the control group who only received routine prenatal care. Psychological symptomology, including relational sensitivity, animosity, anxiety, and paranoia were also significantly lower in the mothers in the therapy group 6 weeks post-delivery.

### 3.2. Studies Investigating the Effects of Music-Based Interventions during Caesarean Sections

#### 3.2.1. Studies That Used Participant-Selected Music during Caesarean Sections

Kaur et al. [54] assessed the effects of participant-selected music played during the C-section on mothers’ anxiety levels using subjective and objective measures of anxiety in 60 participants. The control group did not listen to music but still wore headphones. No differences in serum cortisol levels were found before and after surgery in the music group. However, in the control group, there was a rise in cortisol levels following surgery, suggesting that music may be beneficial in preventing excessive stress in mothers. Music also reduced participant-rated feelings of anxiety from pre- to post-surgery. There were no significant differences in cardiovascular parameters, including HR, SBP, and DBP in the music group.

In another study of 60 participants, those in the experimental group listened to their favourite songs throughout the C-section, whereas those in the control group received standard care with no music [55]. The analysis revealed that participants in the music-intervention group experienced positive changes post-operation, including decreased anxiety, SBP, and DBP, improved oxygen saturation, and lower HR. The control group also exhibited some positive changes, including a reduction in body temperature and DBP. However, participants’ HRs increased in the control group and decreased in the experimental group post-surgery.

#### 3.2.2. Studies That Examined the Use of Participant-Selected Prespecified Music during Caesarean Sections

A large study consisting of 305 participants researched the effects of participant-selected music played during C-sections on mothers’ stress and anxiety levels [56]. The findings showed that mothers who listened to music experienced significantly lower anxiety and had reduced cortisol levels, SBP, and HR during specific stages of the C-section compared with mothers in the control group who received routine care with no music intervention during surgery. Amylase and DBP levels did not vary significantly between both groups.

Chang and Chen [57] were interested in the outcomes of different types of anxiety-relieving music played during caesarean deliveries on 64 Taiwanese mothers’ anxiety and satisfaction levels. The prespecified music selected by the study participants was shown to have anxiety-alleviating effects when scores were compared against those in the control group, which received routine nursing care and regular communication with the researcher but no music was played. Similarly, satisfaction throughout the C-section was rated significantly higher in the music group than in the control group.

Another study of 50 participants examined the effects of participant-selected prespecified music played before and after caesarean delivery on mothers’ anxiety levels [58]. In this study, music included classical, pop/top 40, R&B, country, soft rock, and gospel; they did not significantly reduce state-anxiety. However, when compared with the control group, which received standard preoperative care with no music, the music group experienced less variation in anxiety scores recorded pre-and-post-C-sections, suggesting that preoperative music may be beneficial in preventing wavering anxiety levels. Participants strongly favoured incorporating music in future birthing experiences.

One RCT explored the impact of music played during C-section on HRV, anxiety levels, and pain scores for 60 mothers [13]. The findings revealed that participant-selected Chinese classical music played 30 min prior to the procedure resulted in lower anxiety scores post-C-section. Contrastingly, anxiety scores in the control group did not vary pre- and post-surgery. Similarly, pain scores in the experimental group were significantly lower six hours post-delivery.

#### 3.2.3. Studies That Used Researcher Preselected Music during Caesarean Sections

A study of 49 participants found that those who listened to Sufi music during the operation had significantly lower anxiety levels compared with those in the control group [59]. Pre-and-post-surgical vital signs, including SBP, DBP, and oxygen saturation levels (OSL) in the experimental group were unaffected. However, HRs and RRs in the music group were significantly lower post-C-section. Vital signs in the control group were similar pre-and-post-surgery.

Kurdi and Gasti [60] assessed the outcomes of two variations of meditation music during caesarean delivery on the anxiety levels, pain perception, and psychological well-being of mothers post-surgery. Involving a cohort of 189 participants, the study compared the effects of meditation music with a control group that listened to a blank MP3 player. The outcomes revealed that calming and binaural beat meditation music was notably successful at decreasing anxiety and pain up to 24 h post-delivery.

Another study of 105 participants examined two presurgical therapies, i.e., MT, namely the composition “weightless” by Macaroni Union and Benson’s relaxation technique (BRT) [14]. Both interventions significantly reduced anxiety levels during caesarean deliveries compared with the control group receiving standard nursing care, although BRT was more effective.

#### 3.2.4. Studies That Used a Combination of Participant-Selected and Participant-Selected Prespecified Music during Caesarean Sections

One study examined the effects of playing participant-selected music or, where participants had no favourite songs, music chosen from a prespecified list during the perioperative period on postoperative pain and cardiopulmonary parameters in 60 women [61]. The findings showed that listening to music reduced the mothers’ HRs, RRs, and pain scores and increased the time interval before pain-relieving medication was requested.

## 4. Discussion

### 4.1. Summary of Results

To the best of our knowledge, this is the first systematic review and narrative synthesis to explore the effectiveness of music-based interventions during vaginal labour and caesarean delivery using only RCTs. The literature search generated 28 studies, with a sum of 3835 participants included in the final evaluation. The extracted studies were grouped into two broad categories: studies examining the effects of music-based interventions during vaginal labour and studies investigating the effects of music-based interventions during caesarean section.

In summary, this review showed that most but not all the included studies found that music was beneficial in reducing pain and anxiety during vaginal labour [3,27,38,39,40,41,42,43,44,45,46,48,49,50,51,52,53] and C-section [13,14,56,57,59,60,61], particularly in primiparous women. However, other studies did not find a significant advantage of music over a control condition or a different therapy [47,58,60].

The application of music as one element of a larger treatment during vaginal labour [27,53], as individually selected music [41,52,54,55], as music combined with another therapy [27,51,52,53], as instrumental [3,14,40,43,46,48], classical [13,42,53] and relaxing [27,42,45,50,51,57,59,60] styles of music, and as music played via headphones during the caesarean procedure [14,54,57,59,60,61] all seemed particularly helpful in reducing pain and anxiety in mothers.

However, some studies revealed that the analgesic effects of music diminished as labour progressed [3,38]. Two studies found that music did not improve anxiety post C-section [5,54] and music, whilst effective during labour, may lead to increased anxiety 2 days postpartum [41]. Although music was effective at relieving pain and anxiety, massage [47,49], acupressure [51], and a combination of dance and music [52] were found to be more efficacious.

Thus, overall, music seems to be beneficial in reducing pain and anxiety during vaginal labour and C-section. It appears that this therapeutic effect can be amplified by combining music with massage, acupressure, or dance. However, the long-term effects of music during childbirth and C-section are not clear.

### 4.2. Comparison with the Results of Previous Systematic Reviews

The results of our review are mostly consistent with findings from previous systematic reviews that suggest that music is an effective method of pain and anxiety relief during childbirth.

Previous systematic reviews reported meta-analyses evaluating the impact of music on pain and anxiety during labour; however, they faced challenges stemming from the heterogeneity in methodologies in music intervention studies. In a systematic review by Chuang et al. [28], a meta-analysis was conducted to evaluate the effects of music on pain and anxiety management during labour. While individual studies in the review indicated that music played during childbirth alleviated pain, the aggregation of results revealed considerable heterogeneity and a lack of statistical significance. Given these challenges and the variances in methodologies and outcomes in the existing literature, a narrative synthesis seemed a more appropriate approach for our study. This method of analysis allowed for a nuanced examination of the various methodologies and outcomes, ultimately yielding valuable insights beyond the scope of a traditional meta-analysis.

The results of our study indicated that the pain-alleviating effects of music during vaginal labour became less effective as labour progressed. This finding contrasts with a prior systematic review by Santiváñez-Acosta et al. [29], which assessed the effects of music on pain and anxiety management in primiparous women during labour and found that music alleviated pain throughout the latent and active phases of labour.

Our study also extended the findings of Weingarten et al.’s systematic review and meta-analysis, which examined the effects of music played during caesarean delivery on the mothers’ anxiety levels [31]. Both our study and Weingarten et al.’s study recognized the positive impact of music played during the intraoperative period. However, our research further emphasized the specific advantage of using headphones in this context.

Another systematic review by Lin et al. [30] included studies that assessed the anxiety-relieving effects of music in women undergoing a caesarean delivery or vaginal labour. The authors identified a general reduction in anxiety rates within the intervention group; however, they did not categorize the data based on the mode of delivery. Our study categorized and analysed the studies based on the mode of childbirth and therefore demonstrates a methodological improvement as it allows for a more accurate and contextually relevant assessment of the impact of music interventions on anxiety during childbirth, which may be of significant importance for healthcare practitioners and decision-makers.

### 4.3. Limitations of Included Studies

The heterogeneity in methodologies in the included studies makes it challenging to formulate overarching conclusions regarding the effects of music on pain and anxiety during childbirth.

One drawback is the inherent bias in studies employing music-based interventions resulting from the inability to blind participants and researchers to the condition to which they are assigned. Moreover, assessing the effectiveness of music interventions can prove difficult due to the multifaceted nature of music. Some studies did not follow the guidelines for reporting the type of music correctly as advised in the reporting guidelines for music-based interventions [62]. For example, some studies did not state the specific genre/songs or did not give their reasoning for the music chosen. Some participants wore headphones [3,14,40,48,49,50,52,54,57,58,59,60] whilst others did not [41,43,44,45,55,56], which can impact the listening experience, and MT was delivered by a trained music therapist in only two studies [27,45]. Previous research has emphasised the importance of a trained music therapist delivering music interventions for optimal effectiveness [63] as the lack of reported detail and variability in the quality of those delivering music interventions to participants may restrict replicability and the potential of implementing the outcomes in clinical practice. However, under real-world conditions, music is available almost everywhere at no additional costs, whereas music therapists are not always available.

Another limitation is that most of these studies did not assess participants’ music-listening habits or preferences prior to the interventions. What may be relaxing for one individual can be distracting/agitating for another, thus participants may not experience the intended pain-and-anxiety-relieving benefits of the intervention. Further, the birth and delivery processes are very dynamic and can result in the needs of the mother changing from needing music that is relaxing to music that may serve as a distraction from pain, discomfort, and anxiety. This can result in the patient wanting and needing different music for the different stages of birth and delivery.

Furthermore, many of the studies included in the analysis had small sample sizes, and none compared the pain-and-anxiety-alleviating effects of different genres of music. These factors increase the risk of overgeneralising the findings and leave gaps in our understanding.

None of the included studies measured/reported on the potential side effects of listening to music during childbirth. A previous study reported that music can elicit negative emotions and memories [64], which may amplify the pain and anxiety felt by participants. Music can also induce an ‘earworm’ i.e., involuntary repetition of music in an individual’s mind. This phenomenon can be distracting and distressing for the individual [65] and thus acts as a confounding variable in the included studies.

Additionally, most of the included studies used subjective ratings of pain and anxiety. Due to the intensity of labour, some individuals may find it challenging to accurately articulate their pain and anxiety levels, especially during peak moments of distress. Furthermore, trying to recall pain and anxiety post-delivery may result in inaccurate assessments and recall bias (Niven et al., 2000), consequently affecting the validity of findings. Two studies assessed the effects of music as part of a larger intervention [27,53]; however, it is unclear whether it was the music that was effective, or whether it was the additional components of the therapy that led to a reduction in anxiety and pain scores.

None of the included studies reported any long-term follow-ups, for example, one year post-childbirth. This information could be significant in steering the development of more efficacious and targeted music interventions going forward. Furthermore, no study compared the effect of music with the effect of medication during childbirth, and similarly, none of the included studies measured the effects of music on the accompanying person, who is often the father. The birthing partner is crucial as they can have a relaxing and positive influence on the mother, or they might panic and cause additional stress for the mother and professionals during childbirth.

### 4.4. Strengths and Limitations of the Current Review

This is the first systematic review to conduct a narrative synthesis to assess the effects of music-based interventions on both vaginal and caesarean deliveries exclusively using high-quality evidence from RCTs. Moreover, this systematic review analysed the effects of various methods of music selection, i.e., participant-selected, researcher-pre-selected, and participant-selected prespecified music. This approach offers a comprehensive insight into how various methods of music selection can impact outcomes, thus increasing the richness of the findings.

This review had several limitations. First, although extensive measures were taken to uphold academic excellence, the subjective nature of a narrative synthesis meant that we were unable to quantify our data and draw precise conclusions. As there were substantial heterogeneities between studies based on the study design, the group of study participants, the music intervention, and the outcome parameters, a meta-analysis was not possible. Moreover, most of the included studies were conducted in Europe [39,40,41,42,43,44,46,47,52,53,55,56,59] and Asia [3,13,14,27,38,49,50,51,57,61]. No study assessed the effect of music during labour in Africa or South America, therefore the findings of this review may lack external validity.

We planned to include only studies published in English or German. This approach might have increased the language bias and may have impacted the generalisability of the findings. In fact, there was no German article that fulfilled all the inclusion criteria. The only study from Germany included in this systematic review (Hepp et al., 2018) was published in English. There are several potential reasons for this, for example, most German studies on music therapy do not report RCTs, and many German journals that would report music therapy studies, e.g., the Musiktherapeutische Umschau, are not included in any of the three databases we used.

Another limitation is that no manual searching was undertaken. Relying exclusively on electronic database searches can result in overlooking relevant studies that are not indexed or easily obtainable online.

The protocol for this systematic review was not registered on PROSPERO, even though this would have increased the transparency of this work.

We did not differentiate between the therapeutic application of music facilitated by any clinician and music therapy facilitated by a professional music therapist. In Table 1, we used the wording from the original manuscript and did not make judgements about whether the music intervention fulfilled specific criteria that would justify the use of the term “music therapy”.

### 4.5. Implications and Future Directions

The results of this systematic review demonstrate that music-based interventions provide therapeutic benefits for pain and anxiety management during childbirth. It is therefore advisable that midwives and neonatal nurses consider incorporating music into the birthing process due to the psychological and physiological benefits for mothers. It might also be advisable that they consult a trained music therapist, when possible, to ensure appropriate and safe implementation of music-based interventions.

Building on insights gained from this systematic review, future research should consider studying the effects of specific music genres on pain and anxiety in mothers during childbirth. Different music genres can induce varying emotional/physiological responses. This nuanced approach would allow researchers to better understand the influence of music on maternal anxiety and pain perception during labour, ultimately enabling the development of tailored, culturally relevant interventions for mothers.

Additionally, no study has compared the effects/side effects of music with the effects/side effects of medication, including benzodiazepines or pain medication. Whilst it may be that acute pain and anxiety management is not possible and that music could potentially serve as a preventative means of mitigating severe pain and anxiety when introduced early, these assumptions need rigorous empirical validation.

Additional areas for the use of music during the perinatal period might include the effects of music or MT on potentially traumatic experiences of birth [66].

Future research should adhere to reporting guidelines for music-based interventions [62]. This methodology would facilitate more robust evaluations of music-based interventions, ultimately contributing to the execution of higher-quality systematic reviews with reduced heterogeneity. Long-term follow-ups and assessments of potential side effects of music are also recommended in future studies to strengthen the evidence supporting music-based interventions on pain and anxiety in mothers during labour. It would also be useful to investigate the added effect and health-economic value of a trained music therapist.

Finally, future research should also consider assessing the effects of music-based interventions in specific birth settings such as home births, hospital births, and birthing centres.

## 5. Conclusions

The current systematic review is the first to narratively synthesise the use of music-based interventions as a method of pain and anxiety relief for mothers during vaginal and caesarean deliveries. This review builds upon previous studies, illustrating that music interventions can alleviate pain and anxiety during childbirth and lead to improvements in physiological factors, including HR and BP. Participant-selected music, instrumental/relaxing styles of music, and music as part of larger interventions/combined with another non-pharmacological therapy appeared particularly useful. However, the findings suggest that the therapeutic benefit of music might apply primarily to alleviating low-level pain rather than acute pain. Additionally, to ensure the efficacy and safety of these interventions in clinical practice, further research is needed to assess the long-term effects and potential side effects of such interventions in the obstetric setting, along with implementing more rigorous methodologies such as following reporting guidelines for music interventions [62] and enlisting trained music therapists for intervention delivery.

## Figures and Tables

**Figure 1 ijerph-20-07120-f001:**
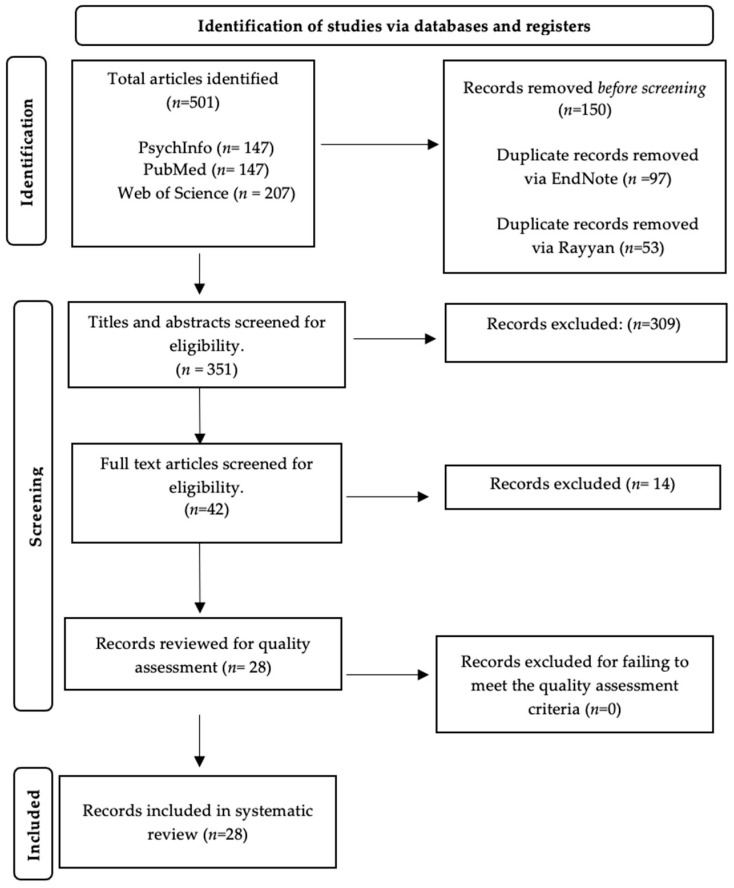
PRISMA Flow Diagram.

**Table 1 ijerph-20-07120-t001:** Detailed summary of included studies.

Authors/Years	Country	Sample/Group Size	Total Sample Size	Mean Age	Measures	Music Intervention	Control Group	Method of Childbirth	Type of Music	Main Outcomes and Statistical Significance
**(1) Studies Examining the Effects of Music-based Interventions During Vaginal Labour.**										
**(1.1) Music-based Interventions as the Experimental Group Compared with a Control Group.**										
Phumdoung and Good (2003) [3]	Southern Thailand	- Primiparous women. Four women were from small community hospitals and 140 women were from a big health centre in Thailand. - Music group (*n* = 55) - Control group (*n* = 55)	*N* = 110	24	VAS	The Music group listened to soft music with earphones during the first 3 h of the active phase of labour.	Traditional labour; however, to avoid demoralization, the control group was told they would receive music at some point (once pain measurements were complete).	Vaginal delivery (active phase).	Choice of five types of Western music without lyrics: synthesizer, harp, piano, orchestra, and jazz.	Participants who listened to soft music experienced less sensation of pain (*p* < 0.01) and pain-related distress (*p* < 0.001) compared with participants in the control group.
Liu et al. (2010) [38]	China	- Primiparous women were predicted to have a normal, spontaneous birth. - Music group (*n* = 30) - Control group (*n* = 30)	*N* = 60	27.12	VASP; PBI; FT; OEQ.	Participants listened to music for 30 min during the latent and active phases of labour. Participants could choose whether or not to wear headphones.	Participants received routine care post-admission. Participants were unaware of the option to listen to music.	Vaginal delivery (latent and active phases).	Participants chose either relaxing music, light music, popular music, crystal children’s, or Chinese religious music.	Mothers who listened to music during labour experienced less pain (VASP, *p* < 0.001) and anxiety (VASA, *p* < 0.001; PBI, *p* = 0.014) and had a higher FT (*p* = 0.014) compared with participants in the control group.
Simavli et al. [39]	Turkey	- Primiparous women who had reached a gestational age of 38. - Music group (*n* = 80) - Control group (*n* = 81)	*N* = 161	Music: 24.17 Control: 23.28	VASA; VASP; VASS; EPDS.	During labour, participants’ selected music was played at all times, with a 20 min break after every 2 h of play. Music continued up to the end of the third phase of labour. Participants could choose whether or not to wear headphones.	Participants received standard prenatal care.	Vaginal delivery.	Six types, including classical music, light music, popular music, Turkish art music, Turkish folk music, and Turkish Sufi music.	Mothers in the music group had lower levels of postpartum pain compared with controls (*p* < 0.001). They also had higher satisfaction with their birthing experience at 2 h (*p* < 0.001), 12 h, and 24 h (*p* < 0.01) postpartum.
Surucu et al. (2018) [40]	Turkey	- Healthy, pregnant primiparous women. - Music group (*n =* 25) - Control group (*n =* 25)	*N =* 50	Music: 22.08 Control: 21.04	VAS; STAI; FAS.	When in active labour (4 cm), women listened to Acemasiran-style music through headphones for 3 h (20 min listening, 10 min not listening).	Routine procedures typically used in hospitals during labour, with no additional practices or music therapy.	Vaginal delivery (active phase).	Acemasiran.	Women in the music group experienced less pain (*p* < 0.05) and lower anxiety levels (*p* < 0.05). Labour was rated as easier by women in the music group (*p* < 0.05), but the contraction period was longer for women in the music group compared with controls (*p* < 0.05).
Buglione et al. (2020) [41]	Italy	- Nulliparous women at full-term with singleton pregnancies. - Music group (*n =* 15) - Control group (*n =* 15)	*N =* 30	Music: 28.7 Control: 31.1	VASP; VASA.	Women listened to music via speakers throughout their labour and the birth of the baby. Women chose songs to listen to.	Participants received the same obstetrical care during labour/delivery as those in the intervention group, except they did not listen to music during labour/delivery.	Vaginal delivery (active phase).	Participants’ own choice of music.	During the active phase of labour, significantly lower levels of pain were experienced in the music group compared with the control group (*p* < 0.01).
Simavli et al. (2014) [42]	Turkey	- Primiparous women who had reached 37–41 weeks of gestation. - Music group (*n =* 77) - Control group *(n =* 79)	*N =* 156	Music: 25.06 Control: 25.09	SBP; DBP; VASP; VASA.	Music was played during the latent phase and the first 2 h of the active phase of labour. 20 min break after every hour of music. Near the end of the active phase and during the second stage of labour, the music type and volume were changed to a more rhythmic beat for pushing.	Participants were provided with a blank CD with no music during labour. The control group only received routine obstetrical care without any additional intervention.	Vaginal delivery (latent and active phase).	Participants were advised to choose soft, relaxing music. Five types were used: classical music, Turkish art music, Turkish folk music, Turkish classical music, and pop music.	Throughout labour and postpartum, pain and anxiety were reported to be lower in the music group compared with the control group (*p* < 0.001).
Garcia-González et al. (2018) [43]	Spain	- Nulliparous women who were going to their routine prenatal care. - Music group (*n =* 204) - Control group (*n =* 205)	*N =* 209	Music: 31.85 Control: 30.90	Childbirth parameters	Prior to birth, participants listened to music via a CD player in a quiet room at home. There was a total of 14 sessions, each lasting 40 min. Music had to be listened to at the same time each day.	Participants went through the same assessment and procedures as the experimental group but without music intervention.	Vaginal labour or Caesarean section.	Instrumental piece of music called “*musical journey through pregnancy*” By Gabriel F. Federico.	Women in the music group had more spontaneous and less induced labour compared with women in the control group (*p* = 0.006).
Amanak (2020) [44]	Turkey	- Primiparae pregnant women aged 20–37. - Music group (*n =* 15) - Control *(n =* 15)	*N =* 30	Music: 25.36 Control: 25.1	IDF; LTF; VAS; STAI–SAI.	Post baseline data collection, participants listened to 30 min of music via a CD player followed by 30 min intermission (conducted in rounds).	The control group had standard care with no intervention.	Vaginal delivery.	Turkish music: instrumental Ney music played in the modal rhythm of Segah.	Labour duration was significantly shorter in the Ney music group compared with the control group (*p* ≤ 0.05). Ney music decreased anxiety and pain.
Browning (2001) [45]	N/A	- Primiparous women with a planned hospital birth. - Music (*n =* 10) - Control (*n =* 10)	*N =* 20	N/A	LAS-E; MPS; TOMRI; Amount/frequency of medication requested; qualitative interviews.	Participants worked with a music therapist to choose music that was soothing/relaxing. A cassette player was used to play music during labour. Sony Walkman was provided if participants wanted to go outside of the birthing room during labour. Six 90-min tapes of recorded music were provided.	Participants in the control group had the same treatment as those in the music intervention group and had a coach but no MT before labour.	Vaginal delivery.	Selection of anxiety-relieving and rhythmic music.	The music group demonstrated a significant increase in perceptions of control according to LAE-S scores (*p* = 0.01). The music group showed significantly higher levels of relaxation compared with the control group (*p* = 0.04).
Garcia-Gonzalez et al. (2018) [46]	Spain	- Pregnant women in their third trimester of pregnancy. - music (*n =* 204) - control (*n* = 205)	*N =* 409	Music: 31.85 Control: 30.90	STAI; Childbirth parameters; birth size.	Music was to be listened to in a quiet room at home over the course of 14 sessions (each session was 40 min) three times weekly at a consistent time.	Participants underwent the same procedure as the experimental group but with no music intervention.	Vaginal delivery or caesarean section.	“*Musical journey through pregnancy*” by Gabriel F. Federico was played on a CD.	Pregnant women who received MT experienced lower levels of anxiety compared with those in the control group (*p* < 0.001). The MT group also had lower anxiety levels (*p* < 0.001).
**(1.2) Music-based Interventions Compared with Other Therapies.**										
Kimber et al. (2008) [47]	England	- Women scheduled for care and delivery at the birthing unit during the study period. - Massage with relaxation *(n =* 30) - Music with relaxation *(n =* 30) - Control group *(n =* 30)	*N =* 90	Massage with relaxation: 30 Music with relaxation: 28 Control group: 30	VAS; customised version of CBWS; LAS.	Participants were trained to focus on breathing and visualisation techniques. Music was used in place of massage during labour.	No music or massage intervention was used; however, participants were encouraged to attend standard antenatal classes running at the site of the trial.	Vaginal delivery.	The mother and her birthing partner picked their favourite music to play during labour.	No significant differences were observed between the massage, music, and control groups.
Estrella-Juarez et al. (2023) [48]	New Zealand	- Nulliparous women at full term (over 37 weeks gestation). - Music intervention group (*n =* 109) - VR group *(n =* 130). - Control group *(n =* 124)	*N =* 343	Music: 32.38 VR: 31.10 Control: 31.60	Childbirth parameters.	Women in the music intervention group listened to music via iPods, with overhead headphones worn, in 20-minute intervals during the first stage of labour.	Participants received the same level of support during the initial phase of labour as the other groups; however, no MT was administered.	Vagina delivery, instrumental, or C-section.	Instrumental piece of music called “*musical journey through pregnancy*” By Gabriel F. Federico.	Significant reduction in the length of the first stage of labour (*p* = 0.008), fewer episiotomies (*p* = 0.04), and fewer non-reassuring FHR tracings (*p* = 0.004) in the music intervention group compared with the control group.
Taghinejad et al. (2010) [49]	Iran	- Participants were primiparous singletons hospitalised for vaginal delivery. - Massage group (*n =* 51) - Music group *(n =* 50)	*N =* 101	Massage: 21.3 Music: 21.5	VAS.	When the woman’s cervix was 3–4 cm dilated (early active phase of labour), mothers listened to soft traditional music without lyrics, whilst wearing headphones, for half an hour.	Participants had no MT but received standard care of pain relief during the birth.	Vaginal delivery (active phase).	Participants had five types of soft traditional music to choose from. (Exact types of music not specified).	Massage and music therapy were both effective, but massage therapy was more effective than music at reducing subjective pain (*p* = 0.011).
Dehcheshmeh and Rafiei (2015) [50]	Iran	- Primiparous women who have reached full term in their pregnancy. - Music intervention group (*n =* 37) - Ice massage group *(n =* 37) - Control group *(n =* 38)	*N =* 112	Music intervention: 21.43 Ice massage: 22.30 Control: 22.90	VAS.	Participants listened to music via headphones for half an hour in a selected room.	Participants received standard labour care without any additional intervention.	Vaginal delivery.	In line with participant preferences, women could listen to either piano music or wave sounds.	Participants in the music intervention group showed significant decreases in pain intensity at three time points i.e., 4 cm dilation, 6 cm dilation, and 8 cm dilation (*p* = 0.001). The music intervention group experienced lower pain levels at all three time points compared with the control group (*p* < 0.05). The average total pain intensity scores were lower in the music group.
Wan and Wen (2018) [51]	China	- Pregnant women who were set to give birth in West China women’s hospital. - Acupressure group (*n =* 60) - Music group (*n =* 60) - Combination *(n =* 62) - Control group (*n =* 59)	*N =* 238	Acupressure: 26.18 Music: 25.57 Combination: 26.70 Control: 26.02	VAS-S; VAS-P; VAS	Music was played continuously for 20 min followed by a 2 h break between sessions.	The control group completed all outcome measures but did not receive any additional interventions.	Vaginal delivery.	Participants were encouraged to choose either relaxing, soft, or regular rhythmic music.	VAS-A anxiety scores in the music group and the group that received music plus acupressure were significantly lower than in the control group (all *p* < 0.05). Pain scores from VAS-P in music, acupressure, and combination groups significantly decreased (all *p* < 0.05) compared with the control group. VAS-P scores in the acupressure group were statistically lower than those in the music group at 1, 4, and 8 h (all *p* < 0.05).
Gönenç and Dikmen (2020) [52]	Turkey	- Nulliparous pregnant women in the active phase of labour. - Dance and music group (*n =* 33) - Music group *(n =* 33) - Control group *(n =* 33)	*N =* 99	Most participants in all three groups were between 18 and 24 years.	VAS; W-DEQA; LMF.	In both the dance and music and the music-only groups, participants were asked to select three songs they liked. Music was played for 30 min via headphones once cervical dilation had reached 4–5 cm.	Participants received standard care involving monitoring of labour progression, vital signs, and foetal heart tones.	Vaginal delivery (Active phase).	A range of upbeat pop music, slow pop music, Turkish folk music, and religious music was selected in both the dance and music group and the music-only group.	Dance plus music and music only reduced labour pain and fear during the active phase of labour (both *p* < 0.001).
**(1.3) Music-based Interventions as Part of a Larger Therapy.**										
Guo et al. (2022) [27]	China	- Primiparous women who underwent a vaginal birth. - Music/free delivery group (*n =* 201) - Traditional delivery group (*n =* 239)	*N =* 440	Music/free delivery: 27.85 Traditional delivery: 27.99	PLPQ; Apgar score; perineal score; amount of bleeding 2 h post birth.	A music therapist personalised the music therapy based on the mothers’ experiences and likings. Different types of music were played during the first, second, and third phases of labour.	Traditional therapy group: standard labour practices were implemented with no music intervention.	Vaginal delivery.	First phase of labour: relaxing/hypnotic music was played. Late stage of the first phase of labour: intense rhythmic music was played. Second/third phase of labour: parent–child music was played.	MT combined with free position during labour resulted in statistically lower levels of pain response (*p* < 0.01) and total pain scores in mothers (*p* < 0.05).
Perkovic et al. (2021) [53]	Bosnia	- Pregnant women in their second or third trimesters. - Music group *(n =* 90) - Control group *(n =* 85)	*N =* 175	Music: 31.4 Control: 30.9	VAS; SCL-90.	Music was part of a group education intervention. It involved participants listening to their choice of classical music for 15 min before bedtime.	Participants received routine prenatal care and usual obstetric practices exclusive of additional interventions	Vaginal delivery.	Classical music	Women who took midwifery education classes and listened to classical music during pregnancy described pain as moderate compared with those in the control group, who rated their pain as severe.
**(2) Studies Investigating the Effect of Music-based Interventions During Caesarean Section.**										
**(2.1) Studies That Used Participant-selected Music During Caesarean Section.**										
Kaur et al. (2023) [54]	N/A	- Pregnant women undergoing elective/planned C-sections. - Music (*n =* 30) - Control (*n =* 30)	*N =* 60	Music: 26.63 Control: 26.17	VAS-A; serum cortisol levels; hemodynamic parameter.	Participants listened to music via headphones for the duration of the C-section under spinal anaesthesia.	No music was played but headphones were still worn.	Caesarean section.	Patient-chosen music (folk, Hindi film music, or religious songs).	In the music group, the levels of serum cortisol post-surgery did not differ from the pre-C-section levels (*p* = 0.583). Anxiety scores in the music group decreased significantly (*p* < 0.001) pre-and-post-surgery. Post-surgery differences in anxiety scores in the music and control groups were statistically significant (*p* = 0.0001).
Eren et al. (2018) [55]	Turkey	- Pregnant women who have had a previous C-section in the last 5 years. - Music group (*n =* 30) - Control group (*n =* 30)	*N =* 60	Music: 30.63 Control: 30.03	VAS; vital findings.	Songs were played via stereo speaker throughout the C-sections.	Standard care was given to participants without MT.	Planned caesarean section.	Participants’ favourite songs.	Statistically significant reductions in body temperature (*p* = 0.00), anxiety score (*p* = 0.022), SBP (*p* = 0.003), and DBP (*p* = 0.011) in both pre-and post-C-section and an increase in OSL (*p* = 0.017) in participants in the music group.
**(2.2)** **Studies That Used Participant-selected Prespecified Music During Caesarean Section.**										
Hepp et al. (2018) [56]	Germany	- Pregnant women who had an indication for primary C-section. - N/A	*N* = 305	33.6	STAI; VAS-A; Salvia samples.	Music was played via CD player as soon as participants entered the operating theatre.	Participants received standard treatment with no music intervention.	Caesarean section.	Women selected their preferred music genre from either classical, jazz, lounge, or meditation music.	The music group displayed significantly less anxiety than the control group (*p* = 0.004). At skin incision, the music group had significantly lower SBP (*p* = 0.002) and HR (*p* = 0.049) compared with controls.
Chang and Chen (2005) [57]	Taiwan	- Women scheduled for a C-section between June and October 2002. - Music group (*n =* 32) - Control (*n =* 32)	*N* = 64	Music: 30.31 Control: 32.31	VASA; Physiological indexes; SCDS	Music was delivered from a portable compact disk player via headphones from the beginning of anaesthesia administration and throughout the C-section.	Participants were unaware of a music option but received standard nursing care and regular communication with the researchers.	Caesarean section.	Participants chose one of the following anxiety-relieving music types: western classical, new-age, or Chinese religious.	Women in the music group experienced less anxiety than women in the control group at the end of maternal contact with the baby during the intraoperative period and post-skin-suture (*p* < 0.01). Participants in the music group also had higher SCDS scores, demonstrating increased satisfaction with the C-section (*p* < 0.01).
Denney et al. (2018) [58]	USA	- Women with routine pregnancies who were scheduled to have a planned C-section. - Music (*n =* 25) - Control (*n =* 25)	*N =* 50	Music: 29.9 Control: 31.2	STAI; Physiological measures; qualitative questionnaire.	Music was played via a portable MP3 player whilst mothers were in the preoperative waiting area and postoperative recovery room.	Participants received standard preoperative care prior to the C-section without any music intervention.	Caesarean section.	Mothers in the music group had to choose a playlist from the following genres: classical, pop/top 40, R&B, country, soft rock, or gospel.	There was no statistically significant difference in state-anxiety scores between the music and control groups. When comparing the music group with the control group, participants talked about their deliveries in more positive ways relative to their previous delivery experiences (*p* = 0.046).
Li and Dong. (2012) [13]	China	- Women undergoing elective caesarean section. - Music group (*n =* 30) - Control group *(n =* 30)	*N* = 60	N/A	SAS; HRV; VAS.	Participants listened to music for half an hour before undergoing surgery and continued to listen to music during the C-Section.	Participants did not receive music intervention; however, they were told to relax for 30 min in a quiet room before having the surgery.	Caesarean section.	Participants chose a piece of classical Chinese music.	Anxiety was significantly decreased after music intervention but not in the control group (*p* < 0.01).
Horasanlı and Demirbas (2022) [59]	Turkey	- Women over 18 years of age with singleton pregnancies over 37 weeks gestation undergoing elective C-section. - Music group *(n =* 26) - Control group *(n =* 23)	*N* = 49	Music: 31.25 Control: 29.64	STAI Vital signs: DBP, SBP, HR, RR, and OSL	Music was played via an earpiece prior to the administration of spinal anaesthesia. Music was played throughout the operation.	Participants did not receive music intervention but did listen to regular atmospheric white noise.	Caesarean section.	Sufi music with a steady rhythm.	Anxiety scores after C-section were significantly lower in the music group than in the control group (*p* < 0.001). No difference in anxiety before and after C-section in the control group (*p* < 0.001).
Kurdi and Gasti (2018) [60]	N/A	- Pregnant women undergoing emergency C-sections whilst under spinal anaesthesia. - Soothing meditation music (group M) (*n* = 63) - Binaural beat meditation music (group B) (*n* = 63) - Control (group C) (*n* = 63)	*N =* 189	Group M: 25.4 Group B: 24.6 Group C: 24.5	VAS-A; VAS-P; PONV; psychological wellbeing questionnaire.	Meditation music was played from an MP3 player via bilateral headphones covering the entire ear, ensuring that any noise interference from the operating theatre was kept to a minimum.	The control group listened to a blank MP3 player via headphones concealing the ear.	Caesarean section.	Group M: calming and soothing meditation music. Group B: Binaural beat meditation music.	Calming/soothing and beat meditation music were effective at reducing postoperative pain and anxiety in comparison to the control group at 6 h (both groups, *p* < 0.05) and 24 h (both groups, *p* < 0.05). However, participants in the music groups required pain-alleviating medication earlier than participants in the control group (both *p* < 0.05).
Abarghoee et al. (2022) [14]	Iran	- Primiparous women scheduled for a C-section in four hospitals in an urban area of Iran. - BRT (*n =* 35) - Music group *(n =* 35) - Control group (*n =* 35)	*N =* 105	BRT: 26.97 Music: 29.97 Control: 29.34	SAI	Women were taken into a private room on their own and Music was played through an MP3 player for 20 min prior to the C-section.	Participants received only standard nursing care that sought to give in-depth explanations of the surgery procedure and recovery.	Caesarean section	The song “Weightless” by Macaroni Union was played as the nonverbal music for the Music group.	Significant reduction in anxiety after both the BRT (*p* < 0.001) and the music intervention compared with the control condition (*p* < 0.001).
**(2.3) Studies That Used a Combination of Participant-selected and Participant-selected Prespecified Music During Caesarean Sections.**										
Halder et al. (2022) [61]	India	- Parturient women undergoing elective C-sections. - Music group (*n =* 30) - control group *(n =* 30)	*N* = 60	Music: 28.4 4 Control: 27.2 4	VAS; VRS; NRS	Participants’ chosen music was played (for 20 min durations) via headphones during the preoperative, intraoperative, and postoperative stages.	Participants in the control group followed the same procedure as those in the experimental group but had no music intervention.	Caesarean section	Participants chose their preferred genre. If participants had no preference, they were asked to choose from vocal i.e., Indian classical, semi-classical, folk, light music, or instrumental i.e., single or multiple musical instruments.	VAS scores of postoperative pain were significantly lower in the music group at 1, 2, and 3 h post-surgery compared with the control group (*p* = 0.0003; *p* = 0.002; *p* = 0.02). Patients in the music group took, on average, 29 min longer than the control group before needing further pain-reducing medication (*p* = 0.017).

Abbreviations: BRT, Benson relaxation technique; CBWS, Cambridge birth worry scale; DBP, diastolic blood pressure; EPDS, Edinburgh postnatal depression scale; FAS, face anxiety scale; FHT, foetal heart rate; FT, finger temperature; HRV, heart rate variability; IDF, individual descriptive form; LAS, Labour agentry scale; LMF, labour monitoring form; LAS-E, attitude towards childbirth scale; MPS, McGill pain scale; LTF, labour tracking form; MT, music therapy; NRS, Numeric rating scale; PBI, present behavioural intensity; PLPQ, perception of labour pain questionnaire; OEQ, open-ended questionnaire; OSL, oxygen saturation level; PONV, postoperative nausea and vomiting; RR, respiratory rate; SAS, self-rating anxiety scale; SAI, state anxiety inventory; SBP, systolic blood pressure; SCDS, satisfaction of caesarean delivery scale; SCL-90, The symptom checklist—90; STAI, state-trait anxiety inventory; STAI–SAI, state-trait anxiety inventory–state anxiety inventory; TOMRI, Trippet objective muscle relaxation inventory; VAS, visual analogue scale; VASA, visual analogue scale for anxiety; VASP, visual analogue scale for pain; VASS, visual analogue scale for satisfaction; VR, virtual reality; VRS, Verbal rating scale; W-DEQA, Wijma delivery expectancy/experience questionnaire.

**Table 2 ijerph-20-07120-t002:** Condensed synopsis of the main findings.

Category of Study	Main Relevant Findings
Music-based interventions as the experimental group compared with a control group.	Most but not all studies found that music led to a reduction in anxiety and pain during labour. Other positive effects of music included a reduction in medication use, improvement in blood pressure and pulse rate, and more spontaneous labours.
Music-based interventions compared with other therapies.	In comparison to other non-pharmacological interventions, music was reported to be an effective intervention for reducing labour pain and anxiety. Massage and VR were slightly more effective than music at lessening pain and anxiety in mothers during vaginal deliveries. Music combined with other therapies such as dance or acupressure was found to be more effective than music alone.
Music-based interventions as part of a larger therapy.	Music-based interventions as part of a larger therapy helped to relieve pain and anxiety in mothers during labour. Combined, these interventions were also shown to lessen maternal haemorrhaging and various postpartum psychological outcomes.
Studies that used participant-selected music during caesarean sections.	The utilization of participant-selected music had consistent positive effects on anxiety reduction.
Studies that examined participant-selected prespecified music during caesarean sections.	Participant-selected prespecified music played exclusively prior to and during caesarean delivery reduced anxiety. Music played before and after surgery (not during) did not reduce anxiety but did prevent anxiety from fluctuating throughout the procedure. Music played during C-section increased maternal birth satisfaction.
Studies that used researcher-pre-selected music during caesarean sections.	Sufi music, meditation music (including calming and binaural beat variations), and instrumental music (specifically the song “weightless by Macaroni Union) lessened anxiety in mothers. Sufi music also decreased post-caesarean pain, HR, and RR.
Studies that used a combination of participant-selected and participant-selected prespecified music during caesarean sections.	Participant-selected music or participant-selected prespecified music reduced HR, RR, and pain scores. Music delayed the interval before the request for pain-relieving medication.

Abbreviations: HR, heart rate; RR, respiratory rate; VR, Virtual reality.

## Data Availability

Not applicable.

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
