# Peer review of "The Effects of Music-Based Interventions for Pain and Anxiety Management during Vaginal Labour and Caesarean Delivery: A Systematic Review and Narrative Synthesis of Randomised Controlled Trials"

_ijerph, 2023, doi:10.3390/ijerph20237120_

Round 1

Reviewer 1 Report

Comments and Suggestions for Authors

Thank you for this interesting manuscript that will be beneficial to the literature. There are just a few small type-o’s to address in the document (especially the use of apostrophes). The following are my other questions and comments:

1.       Lines 22 and 120– music-based interventions and music therapy are “non-invasive”. This very often is true, but not all interventions are considered to be non-invasive; especially if the individual does not like music or does not like the specific type of music being played.

2.       Line 68 – the word “anxious” is used to define “perioperative anxiety”. Would there perhaps be a different way of defining it that does not include the same word in the definition?

3.       Line 120 – recommendation to utilize music therapy during childbirth; however, is this manuscript about music therapy or therapeutic uses of music (music-based interventions)? This just seems a little confusing.

4.       Music therapy is something that is provided by a professionally trained music therapist. If a music therapist isn’t providing the intervention, it is not considered to be music therapy. As mentioned above, it could be a therapeutic use of music or music-based intervention, but not music therapy. Please double check all of the studies listed in Table 1 as it appears that some of the columns refer to music therapy (MT); however, music therapy was not listed as the music intervention. Examples of this include Garcia-Gonzalez et al. 2018) and Estrella-Juarez et al (2023). Another possibility is that the original studies may have misused the term “music therapy”.

5.       Table 1 – excellent description of studies.

6.       Table 2 – excellent table demonstrating main findings.

7.       Excellent list of limitations, implications, and future directions.

Author Response

Response to reviewer 1

  1. Lines 22 and 120– music-based interventions and music therapy are “non-invasive”. This very often is true, but not all interventions are considered to be non-invasive; especially if the individual does not like music or does not like the specific type of music being played.

We thank the reviewer for their excellent comment. We agree that we have to change this. Therefore, we amended the sentence: “Music-based interventions are non-invasive and low-cost with minimal side effects and are increasingly being implemented during the birthing process for pain and anxiety relief.” In the abstract. We are now writing: “Music-based interventions are not physically invasive, they usually have minimal side effects, and they are increasingly being implemented during the birthing process for pain and anxiety relief.”

In section 1.3., we also amended the sentence: “Since music therapy is considered a physically non-invasive, generally cost-effective, natural intervention with limited side effects [8,33], …”

We had already written in section 4.3. of the discussion: “None of the included studies measured/reported on the potential side effects of listening to music during childbirth. A previous study has reported that music can elicit negative emotions and memories [64], which may amplify pain and anxiety felt by participants. Music can also induce ‘earworm’ i.e., involuntary repetitions of music within an individual’s mind. This phenomenon can be distracting and distressing for the individual [65], thus acting as a confounding variable within the included studies.“

  1. Line 68 – the word “anxious” is used to define “perioperative anxiety”. Would there perhaps be a different way of defining it that does not include the same word in the definition?

We agree with the reviewer. The respective amended sentence in section 1.2. now reads: “Perioperative anxiety can be defined as a distressing and unpleasant feeling causing worry and nervousness throughout the preoperative, operative, and post-operative period which creates an emotional reaction to a possible threat.”

  1. Line 120 – recommendation to utilize music therapy during childbirth; however, is this manuscript about music therapy or therapeutic uses of music (music-based interventions)? This just seems a little confusing.

We thank the reviewer for this important comment. We mean both. Therefore, we have amended the respective sentence, and we are now writing: ”Since the therapeutic use of music and music therapy are considered a physically non-invasive, generally cost-effective, natural intervention with limited side effects …”

  1. Music therapy is something that is provided by a professionally trained music therapist. If a music therapist isn’t providing the intervention, it is not considered to be music therapy. As mentioned above, it could be a therapeutic use of music or music-based intervention, but not music therapy. Please double check all of the studies listed in Table 1 as it appears that some of the columns refer to music therapy (MT); however, music therapy was not listed as the music intervention. Examples of this include Garcia-Gonzalez et al. 2018) and Estrella-Juarez et al (2023). Another possibility is that the original studies may have misused the term “music therapy”.

We fully agree with the reviewer. Therefore, we added the sentence “We did not differentiate between the therapeutic application of music facilitated by any clinician and music therapy facilitated by a professional music therapist. In Table 1, we used the wording from the original manuscript and did not make judgements about whether the music intervention fulfilled specific criteria that would justify the use of the term “music therapy”.” to section 4.4. of the discussion.

  1. Table 1 – excellent description of studies.

Thank you for your positive feedback.

  1. Table 2 – excellent table demonstrating main findings.

Thank you. 

  1. Excellent list of limitations, implications, and future directions.

Thank you.

Reviewer 2 Report

Comments and Suggestions for Authors

Dear Author(s),

 The manuscript entitled ‘‘The Effects of Music-based Interventions for Pain and Anxiety  Management During Vaginal Labour and Caesarean Delivery:  A Systematic Review and Narrative Synthesis of Randomised Controlled Trials’’.  Overall, I believe the work to have importance and I think it will be of use to other researchers. I congratulate you all on the exciting manuscript I considered to be well-written. However, I have identified a few minor issues in the text of the manuscript for which I would like to call for your attention. Please find my comments and feedback below with reference to lines.

Abstract

Line 20-22 It is a little vague and not really needed; the second sentence works well as the leading sentence (in my opinion).

Line 35-37 This recommendation is good, but it would be nice to see more specific recommendations for music therapy interventions

1.Introduction

Line 42- 45 Please make sure you cite the original source in your references. For example, in the source you cited number 1, this information belongs to another source.

1.      Raana HN, Fan XN. The effect of acupressure on pain reduction during first stage of labour: A systematic review and meta-622 analysis. Complement Ther Clin Pract 2020; 39:101126.

For the full article, check out this.

2.Materials and Methods

From what I researched, the explanation methods are suitable.

3.Results

The results are self-explanatory, and they seems good.

Table 1 can be added to the supplementary material.

Instead of writing ‘’X and colleagues’’, you can use ‘’et al.’’

4.Discussion and Conclusion

Line 460 no need for this title

More literature can be discussed in line with the results of the research.

Author Response

Response to reviewer 2

Abstract

  1. Line 20-22 It is a little vague and not really needed; the second sentence works well as the leading sentence (in my opinion).

We fully agree with the reviewer. Therefore, we deleted the sentence “Severe pain and anxiety are often experienced during vaginal labour and caesarean delivery and can lead to a range of negative physiological and psychological outcomes in the mother.”

  1. Line 35-37 This recommendation is good, but it would be nice to see more specific recommendations for music therapy interventions.

According to this suggestion, we deleted the sentence: “Further research should follow standardised reporting guidelines from music interventions for better comparability of the findings.” Additionally, we specified our advice as follows: “Music as part of a comprehensive treatment strategy, participant-selected music, music coupled with another therapy and relaxing/instrumental music were specifically useful at reducing light to moderate pain and anxiety.” 

Introduction

  1. Line 42- 45 Please make sure you cite the original source in your references. For example, in the source you cited number 1, this information belongs to another source.
  2. We exchanged reference [1], to “Hutchison, J.; Mahdy, H.; Hutchison, J. Stages of Labor; StatPearls Publishing LLC: San Francisco, CA, USA, 2020.

 and we have checked the whole reference list to include the relevant sources.

Materials and Methods

  1. From what I researched, the explanation methods are suitable.

Thank you.

Results

  1. The results are self-explanatory, and they seems good.

Thank you.

  1. Table 1 can be added to the supplementary material.

Table 1 is the core summary of this systematic review. Therefore, we would prefer to have in the main manuscript. This is also in accordance with the PRISMA guidance for systematic reviews.

  1. Instead of writing ‘’X and colleagues’’, you can use ‘’et al.’’

We completely agree, and we have changed this in the whole manuscript.

Discussion and Conclusion

  1. Line 460 no need for this title

We decided to arrange the discussion according to different sections. This is a structure that we used throughout the manuscript from chapter 1 to 4. Deleting one sub-heading for this specific section would question this overarching structure. Therefore, we would kindly like to ask reviewer 2 to allow us to keep this subheading.

  1. More literature can be discussed in line with the results of the research.

We fully agree with the reviewer that a comparison with previous similar systematic review was missing in our discussion section. Therefore, we added section:

“4.2. Comparison with the results of previous systematic reviews

The results of our review are mostly consistent with findings from previous systematic reviews which suggest music to be an effective method of pain and anxiety relief during childbirth.

Previous systematic reviews have conducted meta-analyses to evaluate the impact of music on pain and anxiety during labour; however, they have faced challenges stemming from the heterogeneity of methodologies within music intervention studies. In a systematic review by Chuang et al. [28] a meta-analysis was conducted to evaluate the effects of music on pain and anxiety management during labour. While individual studies within the review indicated that music played during childbirth alleviated pain, the aggregation of results revealed considerable heterogeneity and a lack of statistical significance. Given these challenges and the variances in methodologies and outcomes within the existing literature, a narrative synthesis seemed a more appropriate approach for our study. This method of analysis allowed for a nuanced examination of the various methodologies and outcomes, ultimately yielding valuable insights beyond the scope of a traditional meta-analysis.

The results of our study indicated that the pain-alleviating effects of music during vaginal labour were less effective as labour progressed. This finding contrasts with a prior systematic review by Santiváñez-Acosta et al. [29], which assessed the effects of music for pain and anxiety management of primiparous women during labour and found that music alleviated pain throughout the latent and active phases of labour.

Our study also extended the findings of Weingarten et al’s systematic review and meta-analysis, which examined the effects of music played during caesarean delivery on the mothers’ anxiety [31]. Both our study and Weingarten et al.'s study recognized the positive impact of music played during the intraoperative period. However, our research further emphasized the specific advantage of using headphones in this context.

Another systematic review by Lin et al [30], included studies which assessed the anxiety relieving effects of music in women undergoing a caesarean delivery or vaginal labour. The authors identified a general reduction in anxiety rates within the intervention group, however they did not categorize the data based on the mode of delivery. Our study categorized and analysed the studies based on the mode of childbirth and therefore demonstrates a methodological improvement, as it allows for more accurate and contextually relevant assessment of the impact of music interventions of anxiety during childbirth which may be of significant importance for healthcare practitioners and decision-makers.”

Reviewer 3 Report

Comments and Suggestions for Authors
  1. Please include search terms e.g., music listening,  maternal, prenatal, antenatal, emergency cesarean, delivery
  2. Please provide the PRISMA checklist as appendix. Was the study protocol registered on PROSPERO?
  3. For the German articles, how was the translation process performed and documented?
  4. SIGN quality assessment does not provide breakdown on the risk of bias. Kindly use a risk of bias graph to show the different types of bias for each study (e.g. selection, performance, attrition etc.)
  5. There are quite a number of studies that use a common outcome measure of VAS or STAI. Did you attempt to pool these data to look at the effects? 
  6. Please discuss on how the difference in age across the country may influence the anxiety/ pain level.
  7.  
Comments on the Quality of English Language

N.A.

Author Response

Response to reviewer 3

  1. Please include search terms e.g., music listening, maternal, prenatal, antenatal, emergency cesarean, delivery

We thank the reviewer for this suggestion. We have inserted the general search terms in the manuscript in section 2.1.:

“The general search terms were:

  • Birth, labour, childbirth
  • Caesarean birth, caesarean section
  • Music intervention, music therapy

However, we made adaptations for these general search terms according to the requirements of each electronic literature database as explained below.”

We believe that this amendment will clarifies our search method.

  1. Please provide the PRISMA checklist as appendix. Was the study protocol registered on PROSPERO?

We agree that this is important information. We followed the PRISMA checklist and completed it while writing this systematic review. We have uploaded the PRISMA checklist as supplementary table. Additionally, we added into section 2.1. “Supplementary table 2 shows the completed PRISMA checklist for this systematic review.”

We did not register the systematic review on PROSPERO. We have added this information into the limitations section of the discussion (section 4.4.)

“The protocol for this systematic review was not registered on PROSPERO, even though this would have increased the transparency of this work.”

  1. For the German articles, how was the translation process performed and documented?

There was no German article that fulfilled all inclusion criteria. The only article from Germany included into this systematic review (Hepp et al., 2018) had been published in English. There are several potential reasons for this, for example, that most German articles on music therapy do not report RCTs, and many German journals that would report music therapy studies, e.g., the Musiktherapeutische Umschau are not included in any of the three databases we used.

We have included this information in section 4.4.

  1. SIGN quality assessment does not provide breakdown on the risk of bias. Kindly use a risk of bias graph to show the different types of bias for each study (e.g. selection, performance, attrition etc.)

The risk bias is covered in section 1 of the SIGN rating. Supplementary table 1 provides the rating for each question in section 1.

  1. There are quite a number of studies that use a common outcome measure of VAS or STAI. Did you attempt to pool these data to look at the effects? 

We agree with the reviewer, that it is worthwhile considering a meta-analysis if common outcome measures have been used. However, there was also heterogeneity regarding various other aspects of the study design. Therefore, we included the following sentence into the discussion (section 4.4.):

“As there was substantial heterogeneity between studies regarding the study design, the group of study participants, the music intervention and the outcome parameters, a meta-analysis was not possible.”

  1. Please discuss on how the difference in age across the country may influence the anxiety/ pain level.

As studies did not report on the effect of age on the study outcome, we were unable to address this interesting question.

Round 2

Reviewer 3 Report

Comments and Suggestions for Authors

The comments are addressed appropriately. I have no further comments.

Comments on the Quality of English Language

N.A.

Author Response

We thank the reviewer for their favorable assessment of our revised manuscript.